# Pipeline Leakage Detection Using Acoustic Emission and Machine Learning Algorithms

**DOI:** 10.3390/s23063226

**Published:** 2023-03-17

**Authors:** Niamat Ullah, Zahoor Ahmed, Jong-Myon Kim

**Affiliations:** 1Department of Electrical, Electronics, and Computer Engineering, University of Ulsan, Ulsan 44610, Republic of Korea; 2PD Technology Cooperation, Ulsan 44610, Republic of Korea

**Keywords:** acoustic emission, leakage detection, pinhole leak, machine learning, random forest, neural network, decision tree

## Abstract

Pipelines play a significant role in liquid and gas resource distribution. Pipeline leaks, however, result in severe consequences, such as wasted resources, risks to community health, distribution downtime, and economic loss. An efficient autonomous leakage detection system is clearly required. The recent leak diagnosis capability of acoustic emission (AE) technology has been well demonstrated. This article proposes a machine learning-based platform for leakage detection for various pinhole-sized leaks using the AE sensor channel information. Statistical measures, such as kurtosis, skewness, mean value, mean square, root mean square (RMS), peak value, standard deviation, entropy, and frequency spectrum features, were extracted from the AE signal as features to train the machine learning models. An adaptive threshold-based sliding window approach was used to retain the properties of both bursts and continuous-type emissions. First, we collected three AE sensor datasets and extracted 11 time domain and 14 frequency domain features for a one-second window for each AE sensor data category. The measurements and their associated statistics were transformed into feature vectors. Subsequently, these feature data were utilized for training and evaluating supervised machine learning models to detect leaks and pinhole-sized leaks. Several widely known classifiers, such as neural networks, decision trees, random forests, and k-nearest neighbors, were evaluated using the four datasets regarding water and gas leakages at different pressures and pinhole leak sizes. We achieved an exceptional overall classification accuracy of 99%, providing reliable and effective results that are suitable for the implementation of the proposed platform.

## 1. Introduction

Pipelines are the primary mechanical component required for long-distance liquid and gas material distribution and transportation. Pipeline networks, therefore, need unwavering quality, high safety levels, and efficiency. The current status of a pipeline network involves colossal annual leakage rates and the corresponding waste of natural resources. Pipeline leaks may have detrimental effects on the environment, human safety, property, and reputation and additionally lead to financial losses from fines and cleanup expenses. BenSaleh et al. [1] noted that it is well-documented that many countries rely heavily on long-distance oil and water transportation from desalination plants to their intended destinations. Unfortunately, significant quantities of these resources are lost annually due to leaks in the pipelines, with an estimated 60% of water being wasted each year due to pipeline leaks [2]. For pipeline operators to achieve optimal performance, a reliable leak detection system (LDS) is imperative. An influential LDS should be able to detect leaks promptly, provide accurate leak localization, minimize false alarms, be easy to retrofit, function well under various operating conditions, and utilize sensors with high dependability and low maintenance requirements. This article proposes an LDS using acoustic emission (AE) and machine learning (ML) algorithms.

Pipelines, as a means of long-distance distribution and transportation, must meet stringent safety standards and maintain consistent efficiency and quality. However, monitoring these pipelines over long distances poses challenges due to the difficulties in maintaining the infrastructure. Therefore, much research has been conducted to develop robust and reliable methods for detecting spills, explosions, and other anomalies in the pipeline infrastructure. These leaks can result in severe environmental damage, financial loss, and even loss of life [3]. Wang et al. [4] discussed the ever-increasing energy needed to support oil distribution, pipeline distribution topological complexity, and the real-time assessment of distribution network safety. Korlapati et al. [5] surveyed and classified leakage detection into three broad categories: visual inspections, internally determined/computational, and externally based methods and techniques. Fiber optic cable-based leakage detection in pipelines is carried out by laying down the cable alongside the pipes, and changes in strain and temperature are observed. By analyzing Raman scattering, the optical fiber cable can provide a means for measuring the temperature [6]. Although optical fiber cable can monitor changes in temperature in various locations, it fails both in monitoring small leakages and determining the exact location of a leakage [7,8]. Christos et al. [9,10] proposed a low-energy and low-cost wireless sensor-based system to immediately detect leakage in metallic pipes. They monitor the changes in the effects of the vibrational signal appearing due to leakage on the pipeline walls. The leakage detection system is proposed by installing the pressure sensor in the middle of the pipeline segment [11]. It is more sensitive to detecting leaks that emerge far away, such as at the inlet and outlet. Continuous wavelet transform is used to transform the time AE signal into an image and then the convolutional neural network is applied to detect the leakage [12]. In summary, several internal and external monitoring methods have been developed to detect pipeline leaks. These include the use of negative pressure wave techniques [13], techniques based on accelerometer [14], time-domain reflectometry [15], distributed temperature sensing systems [16], acoustic emission technology [17], ultrasonic technology [18], and magnetic flux leakage techniques [19]. Among these, acoustic emission technology has gained significant popularity for its ability to quickly detect leaks, with real-time responses, high sensitivity, and ease of retrofit [20,21]. Research in this area has focused on using pattern recognition and feature extraction techniques to construct leak detection models [22]. Studies have demonstrated the effectiveness of techniques, such as wavelet feature extraction and support vector machine classification, for identifying leaks and using frequency-width characteristics to train leak-detection support vector data-description models from time-domain pipeline signals [23]. Claudia et al. [24] summarized the AE descriptor’s different applications for damage analysis in fiber-reinforced plastics. They analyzed the amplitude, frequency, and cumulative acoustic energy regarding fiber-reinforced plastic damage, crack analysis, and crack propagation.

ML is a subset of artificial intelligence, namely algorithms that improve through previous data records and experience [25,26]. The algorithms focus on building mathematical models using training data or sample data to make decisions or predictions without explicit programming [27]. When properly executed, machine learning can enable tasks to be automated at a breakneck pace. As such, it is critical to integrate real-world data with artificial intelligence in fields that require rapid and precise detection, such as pipeline leak incidents. El-Zahab et al. [28] proposed a system that utilizes accelerometer-based monitoring for pressurized water pipelines. The experimental data were analyzed using three machine learning algorithms: support vector machines, decision trees, and naive Bayes. The proposed system demonstrated its effectiveness in accurately detecting leakage events in pressurized water pipelines. Different machine learning algorithms, such as decision trees, random forests, k-nearest neighbors, and neural networks, have been applied to further enhance detection capabilities for analyzing the collected data. Overall, the above research works performed better for leak detection and size identification. However, there exist some shortcomings. Traditional AE hit features can be extracted from the AE signal by defining a threshold above the level of continuous background noise. However, the predefined threshold for extracting the AE features can lead to false alarms due to noise in the AE signal. Furthermore, defining a threshold above the level of continuous background noise requires human expertise and domain understanding. Additionally, the type of transportation medium will also affect the AE hits in the signal. In order to address the above-mentioned problems, it is of primary importance to develop a leak-sensitive model for pipeline leak detection and size identification. As such, in this manuscript, an attempt has been made to develop a leak-sensitive model for pipeline leak detection and size identification with different transportation mediums, such as fluid and gas. Instead of utilizing a predetermined threshold for AE feature extraction, in this paper, a sliding window is used. In order to exploit the statistical changes in the AE signal due to the defects in the pipeline, statistical indicators, such as kurtosis, skewness, mean value, RMS, peak value, standard deviation, and entropy, are calculated from each sliding window. Furthermore, the changes in the frequency spectrum due to the defect in the pipeline are utilized by calculating spectral features from each sliding window. Additionally, a set of classification models were tested and validated for pipeline leak detection and size identification by considering two different transportation mediums, out of which the best classification model is reported in this study.

The overall novelty and contribution of this work can be summarized as follows:(i)In order to exploit the statistical changes in the AE signal due to the defects in the pipeline, a sliding window is used, and from each sliding window, temporal statistical indicators are calculated. Furthermore, the changes in the frequency spectrum due to the defect in the pipeline are utilized by calculating the spectral features from each sliding window. To the best of our knowledge, utilizing a sliding window to extract statistical and spectral indicators from the AE signal is reported for the first time in this work;(ii)A pipeline health-sensitive classification model is reported in this study based on evaluating different classification models for pipeline leak detection and size identification by considering two different transportation mediums, such as fluid and gas;(iii)Real-world industrial fluid pipeline data were utilized in this study for leak detection and size identification using machine learning algorithms.

Overall, the proposed platform achieved an exceptional overall classification accuracy of 99%, which makes it a reliable and effective solution for pipeline leakage detection and leak pinhole size identification.

The following sections make up the structure of the paper. Section 2 proposes the architecture, methodology, and ML algorithms for leakage detection. The results and pipeline experimental test rig are presented in Section 3. Section 4 presents the conclusion of this study.

## 2. The Proposed Architecture and Methodology

Figure 1 and Figure 2 show the overview of the proposed methodology for pipeline leakage detection. The architecture is implemented using acoustic emission sensors. There are three sensors placed on the pipeline at locations channel 3 = 0 mm, channel 2 = 1600 mm, and channel 1 = 2500 mm. Data are transmitted to the next step, which is the data acquisition step, in which the signals that gauge physical circumstances in the real world and transform the resulting samples into digital numeric values (that a computer can work with) are sampled. The acquisition of data at the defined sampling rate, 1MHz, extracts the desired features and assigns the labels to the feature vector extracted from one second for the complete data set. The next step is to complete the dataset, which is then processed for testing the classification accuracy with different algorithms to detect the leakage in the pipeline. Once the activity labels are identified, the activity gets either “leakage” or “normal” assigned to its label. It will generate the output of the sensor data containing the leakage or the data are “Normal”. The final step is to show the result on the display for the monitoring supervisor. In the rest of the following subsection, we briefly explain each subcomponent.

### 2.1. Acoustic Emission

A leak in the pipeline results in a change in the structural integrity of the material. This change in the structural integrity can be due to fatigue rupture, stress cracks, corrosion cracks, and structural discontinuities. The structural discontinuity or leak in the pipeline (irrespective of the cause) will disturb the flow of the fluid or gas inside the pipeline. However, the intramolecular interactions or chemical bonding of the fluid will force the fluid to keep its flow consistent [29]. Thus, for the fluid to keep its flow consistent inside the pipeline, the molecule of the fluid will exert pressure on the position of pipeline structural discontinuity or leak, which will result in the short, rapid release of energy in the form of an elastic wave. An AE can be defined as a transient sound wave that is generated by a short, rapid release of energy in the form of an elastic wave that is produced by the change in the structural integrity of the material within a specimen, such as a pipeline [30]. The physical phenomenon resulting from the fluid interaction with the structural discontinuity is referred to as an AE event. This AE event is detected by the AE sensors in the form of AE hits. Thus, the AE hits in the signal can be related to a leak due to fatigue rupture, stress cracks, corrosion cracks, and structural discontinuities. For this reason, the nondestructive method based on AE is considered ‘global’ in nature. The word global means that AE-based monitoring allows the investigator to get a bigger picture of the overall performance of the specimen, irrespective of the cause for degradation [31]. Specifically, in this study, a nondestructive method based on AE was used to investigate the health conditions of the pipeline. Based on the global nature of the AE, theoretically, the proposed method will work to identify a leak and its size.

### 2.2. Acoustic Emission Sensor

An acoustic emission (AE) sensor is a device that detects and analyzes the sound waves generated by changes in the internal structure of a material or structure. The R15I-AST, manufactured by MISTRAS Group, Inc, is one example of such a sensor. It uses piezoelectric transducers to convert mechanical stress or strain into electrical signals, which can then be analyzed to determine the location and severity of structural changes. These sensors are commonly used for non-destructive testing and the structural health monitoring of various structures, such as bridges, pipelines, and pressure vessels. With the capability of working under high temperatures, humidity, and pressures, R15I-AST can monitor the structural integrity in near real-time and provide early warnings of potential issues, allowing for the necessary actions to be taken before any damage occurs. The operating frequency range of the R15I-AST sensor and the parameters used for the data acquisition are listed in Table 1.

### 2.3. Data Acquisition

In data acquisition, the physical conditions in the real world are measured through the use of sensors. These measurements are then sampled and translated into digital numeric values that a computer can interpret. This is typically achieved by converting analog waveforms into digital values for further processing. The key component of the current data acquisition system is the acoustic emission sensor, which is used to convert physical parameters into electrical signals. Once the data is acquired from the sensors, it can be used to detect leaks or other anomalies in the pipeline. The spectrum was performed on the time amplitude signal to instantly identify the leak time. Figure 3a–c show the time response for the three sensors, and Figure 3d–f show the corresponding spectrum when there is no leakage. It clearly shows that both the time amplitude and frequency amplitude are low for both time response and frequency response under normal conditions. Figure 3g–i show the time-domain amplitude, and Figure 3j–l show the frequency response. These figures illustrated that when the leakage was introduced, both the time and frequency amplitude increased two times that of the normal condition.

### 2.4. Features Extraction in the Time and Frequency Domains

The feature extraction and selection methods are helpful for the transformation of data, which translates the preprocessed data into processed data to identify significant trends. Features extraction is an essential approach for reducing data size and provides valuable information for developing a classification model. Most researchers use statistical characteristics approaches for feature extraction. Chai et al. [32] extracted the various features from the AE signal, such as peak amplitude, entropy, energy, count, peak frequency, and centroid frequency, to find crack growth under different stress ratios. Muir et al. [33] reviewed the time domain, frequency domain, and composite features extracted from the AE signal for the damage analysis of a structure. They mentioned around 31 various features used in the literature. Traditional AE hit features, such as rise time, decay time, counts, etc., can be extracted from the AE signal by defining a threshold above the level of continuous background noise. However, the predefined threshold for extracting the AE features can lead to false alarms due to noise in the AE signal. Furthermore, defining a threshold above the level of continuous background noise requires human expertise and domain understanding. In order to address this concern, instead of utilizing a predetermined threshold for AE feature extraction, in this paper, a sliding window is used. In order to exploit the statistical changes in the AE signal due to the defects in the pipeline, this research extracts 25 statistical time and frequency domain features for each AE channel using a sliding window. A total of 75 features are extracted from the three AE channels. All these features are provided as input to the classification model for the task of a pipeline health assessment. The features extracted from each AE channel, comprising 11 time domain features (such as mean, standard deviation, skewness, kurtosis, crest factor, clearance factor, etc.) and 14 frequency domain features (namely, P1, P2, P3, P4, P5, …, P14). Considered the frequency spectrum-based feature extraction that incorporated the lower and higher frequencies into their power. Figure 4 shows the details of feature extraction and feature vectors. Table 2 and Table 3 show the mathematical formulas for the time domain and frequency domain features, respectively.

### 2.5. Machine Learning Algorithms for Leakage Detection and Identification

Classification algorithms are used to predict the class or label of a given set of data. The input to these algorithms is a set of features extracted from raw sensor data, which are associated with specific activities or classes. A decision rule or function that can accurately predict the class of new data based on these features must be determined. This process is known as the classification task.

Classifiers are machine learning techniques that can be used to assign labels to activities. They are trained on a dataset of labeled data, where the feature vector (also known as the training dataset) has been given a label. The learning algorithm adjusts its parameters to generate a model or hypothesis, which can then be used to predict the label ‘y’ for new input data ‘x’. In this research paper, the data collected were related to a pipeline, and we used MATLAB to extract the features of each piece of data. Then, we applied algorithms to classify the data using the software tool MATLAB. In order to check the accuracy of the data, four distinct algorithms were used.

A simple and successful technique for classification and regression applications is k-nearest neighbors (KNN). It operates by locating the K closest data points in the training set for a specific point in the test set, then making a prediction using the labels or values of these nearest neighbors. The Euclidean distance, or the distance along a straight line between two points, is one approach to gauge the separation between data points. The Euclidean distance in KNN is determined as the square root of the sum of the squared coordinate differences between the two points. The performance of the KNN model can be influenced by the distance metric that is selected. Euclidean and Manhattan distances can both be employed. In some instances, Euclidean distance may be more appropriate, while the Manhattan distance may be more suitable in others. The dataset’s characteristics and the task when deciding which distance metric to use with KNN.

A random forest is a method of categorization that relies on the construction of many decision trees (weak learners) and, in the end, adopts the verdict reached by the majority of such learners. The decision tree is a single tree, but the random forest has multiple trees. Normally, Overfitting can be prevented through the use of trimming decision trees. With pruning, you have to choose between precision and simplicity. Complexity, extra work, and more use of resources are the results of not trimming. Equal to the parameters of a decision tree classifier is the random forest.

Three different node types are formed while constructing a decision tree, i.e.,

-The root node is the node with no input link and can have no or some output links;-Internal nodes have one input link and two or more output links;-Leaf nodes are the end nodes that have exactly one input link and no output link.

The neural network can be used for more complex models, which can be utilized in multi-class classification. Neural networks are inspired by the brain, which is a network of neurons. The neuron model consists of some inputs with input weights, a hidden layer, and an output (hypothesis). They translate information through a sort of machine recognition, marking, or grouping of the information. The examples they observe are numerical in vector form, into which all correct information, might be pictures, sound, content, or time arrangement, must be deciphered. A neural network is a collection of “neurons” with “synapses” which are connecting. Hidden layers are vital when the neural system needs to realize something truly confounded, relevant, or non-self-evident, like picture acknowledgment. The circles speak to the neurons, and the lines speak to the synapses. Synapses take the input and multiply it by weight. The neurons add the outputs from all synapses and apply an activation function.

### 2.6. Performance Metrics

In order to evaluate the performance of the proposed method in comparison to the reference method, metrics such as accuracy, precision, and recall were employed. Equations (1)–(3) were used to determine these metrics:(1)$$\mathrm{Precision}=\frac{\sum_{a}^{A}n_{a\;\times }\;\left(\frac{{\mathrm{TP}}_{a}}{{\mathrm{RP}}_{a\;+}{\mathrm{FP}}_{a}}\right)}{N}\;$$
(2)$$\mathrm{Recall}=\frac{\sum_{a}^{A}n_{a\;\times }\;\left(\frac{{\mathrm{TP}}_{a}}{{\mathrm{RP}}_{a\;+}{\mathrm{FN}}_{a}}\right)}{N}\;$$
(3)$$\mathrm{Accuracy}=\frac{\sum_{a}^{A}n_{a\;\times }\;\left(\frac{{\mathrm{TP}}_{a}}{{\mathrm{RP}}_{a\;+}{\mathrm{FN}}_{a}}\right)}{N}\;$$

In this context, TP_a_, FP_a_, and FN_a_ refer to the true-positive, false-positive, and false-negative results, respectively, obtained from the features that are representative of class a; n_a_ represents the total number of samples from class a; A represents the overall number of classes in the dataset. The variable N denotes how many samples there are in all of the testing sets.

## 3. Results and Discussion

In this section, the experimental setup and the performance comparison of the machine learning algorithms are described.

### 3.1. Experimental Setup

The proposed technique was experimentally validated using a specific experimental setup, as illustrated in Figure 5. The setup consists of a stainless-steel pipeline with a thickness of 6.02 mm and an outer diameter of 114.3 mm, and it is used for transporting water or gas and is equipped with AE sensors and a data collection system. The sensors used in this setup are R15I-AST-type sensors manufactured by MISTRAS Group, Inc. The sensors were fixed to the pipeline by using adhesive gels and mounting tapes. The data collection system comprised a NI-9223 National Instruments Data Acquisition system and a computer system that was set up to record information on the pipeline data conditions. In order to ensure the sensor’s sensitivity to the applied stresses, the sampling frequency was adjusted to 1 MHz, and pencil lead break tests were carried out.

Leak simulation was conducted by incorporating a valve into the pipeline and conducting tests at pressure levels of 13 and 18 bar. The process involved initially closing the valve and collecting data for 2 min while the pipeline was in normal operation. The valve was then opened to introduce a 1 mm leak, and data were collected for an additional 4 min. When the valve was closed again, the pipeline’s flow was stabilized. This process was repeated at both pressure levels for the same leak pinhole size. In total, 360 signal samples were collected for each test, with 120 samples from the normal condition and 240 samples from the leak condition being used for further analysis and evaluation.

#### Dataset Collection and Description

In this study, the pipeline provides a transportation medium for fluid and gas. For each transportation medium, a pressure of 13 and 18 bar was adjusted with the help of a centrifugal pump. First, the valve was kept closed, and the pipeline was operated to transport fluid under 13 bar pressure; thus, the normal operating conditions data were acquired. After the acquisition of the normal condition data, under the same pressure condition, the leak valve was opened to 1 mm, and data were acquired, which formed Dataset-1. After the acquisition of data under a pressure of 13 bar, the leak valve was kept closed, and the pipeline was operated to transport fluid under a pressure of 18 bar, and the normal conditions data were acquired. After the acquisition of the normal condition data, under the same pressure condition, the leak valve was opened to 0.7 mm, and data were acquired, which formed Dataset-3. The same process was repeated for the acquisition of Dataset-2 and Dataset-4. However, for safety purposes, the leak valve was only opened up to 0.5 mm during the data acquisition for Dataset-2 and Dataset-4. A detailed description of each dataset is given in Table 4.

### 3.2. Performance Comparison of Machine Learning Algorithms for Pipeline Leakage Detection

#### 3.2.1. Neural Network

The dataset was divided into 70% for training, 15% for testing, and 15% for validation, respectively. The ‘logsig’ activation function and two configurations of neurons were used, i.e., 10 and 50 neurons. Figure 6 shows the convergence curves for the best validation value 3.6829 × 10^−7^ at epochs 24 and 1.44345 × 10^−7^ at epochs 28 for 10 and 50 neurons, respectively. The convergence clearly depicts improved accuracy when increasing the number of neurons but also costs an increase in training time. Figure 7 shows the confusion matrices obtained using the 10 and 50 neurons, respectively. The neural networks achieved the highest accuracy for both the 10- and 50-neuron setups for training, testing, and validation. Confusion matrix “1” represents “No Leakage/Normal”, and “2” represents “Leakage”.

Figure 8 shows the convergence curves for the best validation value 2.8079 × 10^−7^ at epochs 24 and 1.5926 × 10^−7^ at epochs 30 for 10 and 50 neurons, respectively. The convergence clearly depicts improved accuracy when increasing the number of neurons but also costs an increase in training time. Figure 9 shows the confusion matrices obtained by using 10 and 50 neurons. The neural networks achieved the highest accuracy for both the 10- and 50-neuron setups for training, testing, and validation.

Figure 10 shows the convergence curves for the best validation value 4.1322 × 10^−7^ at epochs 26 and 1.2969 × 10^−7^ at epochs 31 for 10 and 50 neurons, respectively. The convergence clearly depicts improved accuracy when increasing the number of neurons but also costs an increase in training time. Figure 11 shows the confusion matrices obtained by using the 10- and 50-neuron setups. The neural networks achieved the highest accuracy for both the 10- and 50-neuron setups for training, testing, and validation.

Figure 12 shows the convergence curves for the best validation value 0.00067013 at epochs 20 and 1.5746 × 10^−7^ at epochs 29 for 10 and 50 neurons, respectively. The convergence clearly depicts improved accuracy when increasing the number of neurons but also costs an increase in training time. Figure 13 shows the confusion matrices obtained by using 10 and 50 neurons. The neural networks achieved the highest accuracy for both the 10- and 50-neuron setups for training, testing, and validation.

#### 3.2.2. K-Nearest Neighbor

The datasets were divided into 90–10%, 80–20%, and 70–30%, and used K = 5 and the Manhattan distance as a similarity metric to predict the closest label for the test sample. Figure 14 shows the accuracy of the four datasets; the highest accuracy of 100% was achieved.

#### 3.2.3. Random Forest

The dataset was divided into 90–10%, 80–20%, and 70–30%, and the number of trees = 100, and the Gini index was used for splitting the trees. Figure 15 shows the accuracy of the four datasets; the highest accuracy of more than 99% was achieved.

#### 3.2.4. Decision Tree

The dataset was divided into 90–10%, 80–20%, and 70–30%, and we used the Gini index for splitting the tree. Figure 16 shows the accuracy of the four datasets; the highest accuracy of more than 99% was achieved for each dataset.

#### 3.2.5. The Overall Performance Comparison of the Applied ML Algorithms

Figure 17 shows the overall performance comparison of the applied ML algorithms in terms of prediction accuracy. It shows that the neural networks show the highest accuracy among all the algorithms, and also, the KNN algorithms for all datasets in all splits were the best.

## 4. Conclusions

This article presents a machine learning-based platform for detecting and localizing pipeline leaks using acoustic emission (AE) technology. By extracting various statistical measures from AE signals and using them as features to train machine learning models, the platform can accurately identify and locate leaks in pipelines. In order to preserve the characteristics of both bursts and continuous-type emissions, a sliding window with an adaptive threshold was used, allowing for real-time data collection and analysis. The article also presents an evaluation of the proposed platform by using four datasets that contain water and gas leaks at different pressures and various machine learning classifiers like neural networks, decision tree random forests, and k-nearest neighbors. An overall classification accuracy of 99% was achieved, indicating that the proposed platform is a reliable and effective solution for pipeline leak detection and localization. Overall, the article emphasizes the significance of pipeline leaks and how the proposed machine learning-based platform can be an effective solution for this problem. The severe consequences of pipeline leaks include wasted resources, health risks, distribution downtime, and economic losses, so it is important to develop efficient leak detection systems. The success of the proposed platform in detecting and localizing leaks with high accuracy provides a strong indication of its potential for use in real-world applications. It is also notable that AE technology is a promising solution for detecting pipeline leaks, as it is capable of leak diagnosis, which has been significantly demonstrated. The current study is capable of detecting and identifying the size of a leak in the pipeline. However, the classification model cannot predict the condition of the pipeline, along with the pressure and transportation medium. For this reason, in the future, a classification model can be developed that can predict the condition of the pipeline, along with the pressure and transportation medium.

## Figures and Tables

**Figure 1 sensors-23-03226-f001:**
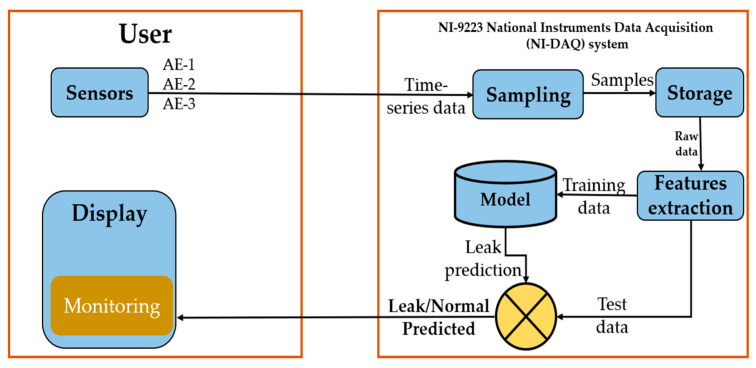
Overview of the proposed methodology for leak detection.

**Figure 2 sensors-23-03226-f002:**
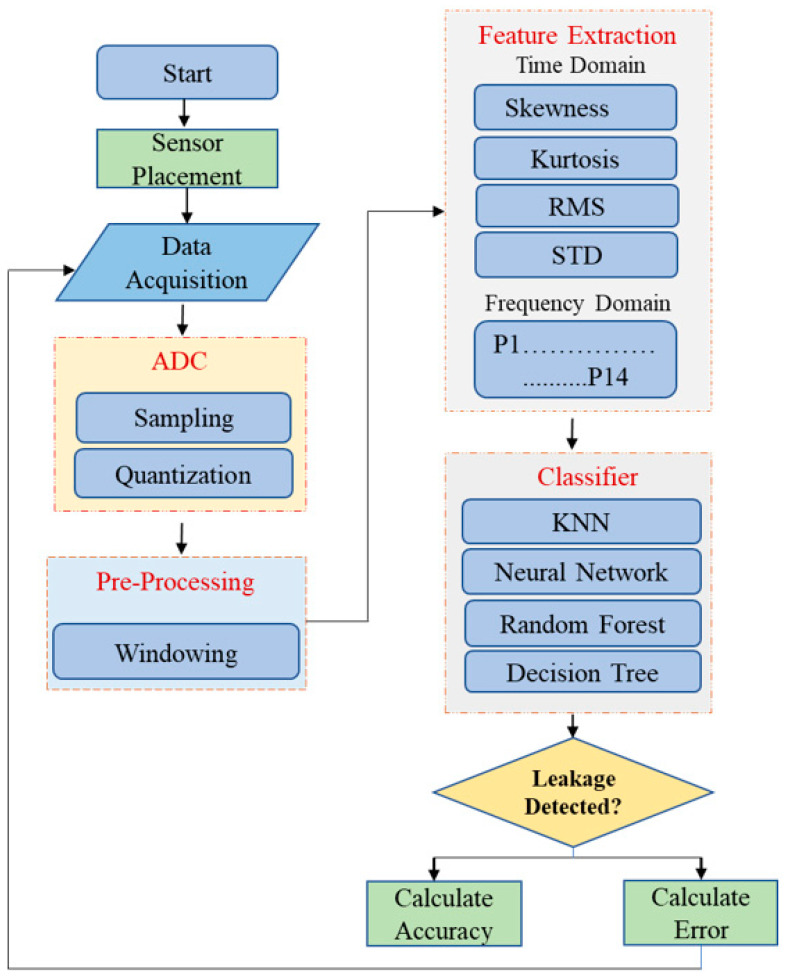
Flowchart of the proposed methodology for leak detection.

**Figure 3 sensors-23-03226-f003:**
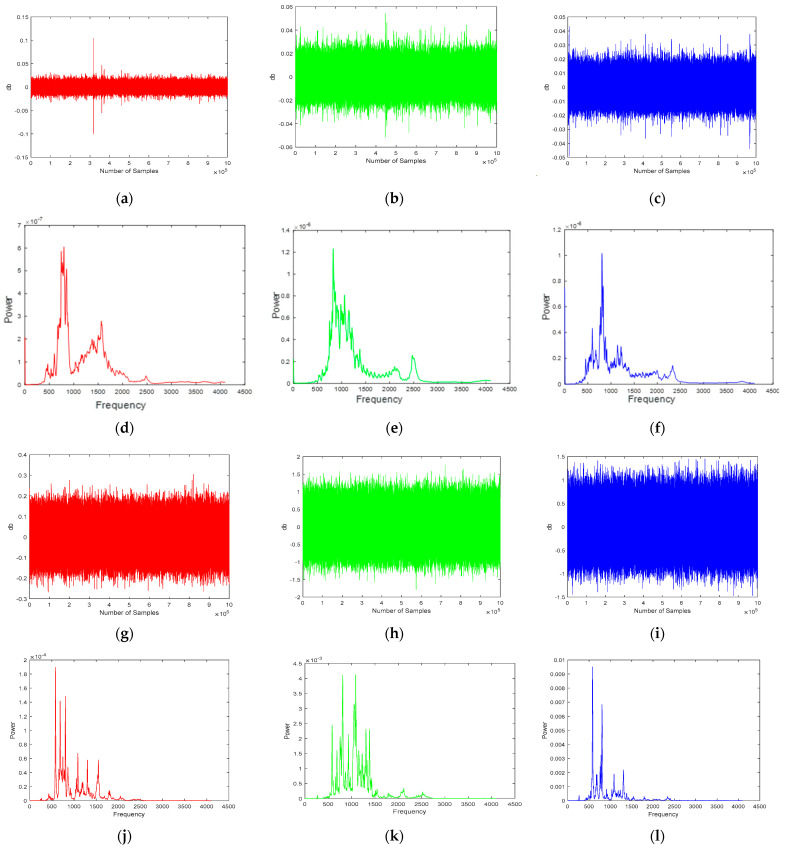
Amplitude and spectral power of the AE signals obtained from the pipeline under normal and leak operating conditions. (**a**–**c**) AE signal under normal operating conditions. (**d**–**f**) Power spectrum under normal operating conditions. (**g**–**i**) AE signal under pipeline leak operating conditions. (**j**–**l**) Power spectrum under pipeline leak operating conditions. (**a**) Channel-1 Time Response No-leakage. (**b**) Channel-2 Time Response No-leakage. (**c**) Channel-3 Time Response No-leakage. (**d**) Channel-1 Frequency Response No-leakage. (**e**) Channel-2 Frequency Response No-leakage. (**f**) Channel-3 Frequency Response No-leakage. (**g**) Channel-1 Time Response Leakage. (**h**) Channel-2 Time Response Leakage. (**i**) Channel-3 Time Response Leakage. (**j**) Channel-1 Frequency Response Leakage. (**k**) Channel-2 Frequency Response Leakage. (**l**) Channel-3 Frequency Response Leakage.

**Figure 4 sensors-23-03226-f004:**
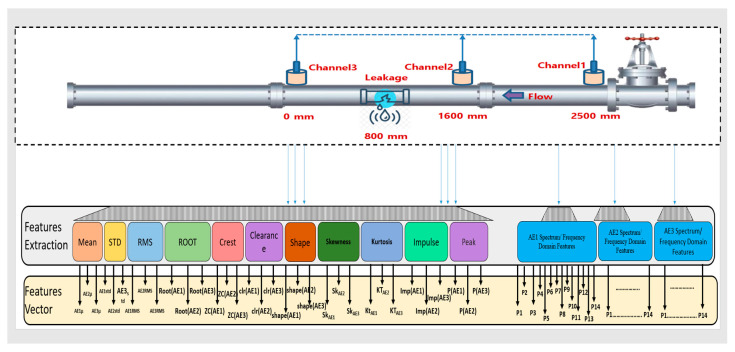
Features extraction and features vector in the time and frequency domains.

**Figure 5 sensors-23-03226-f005:**
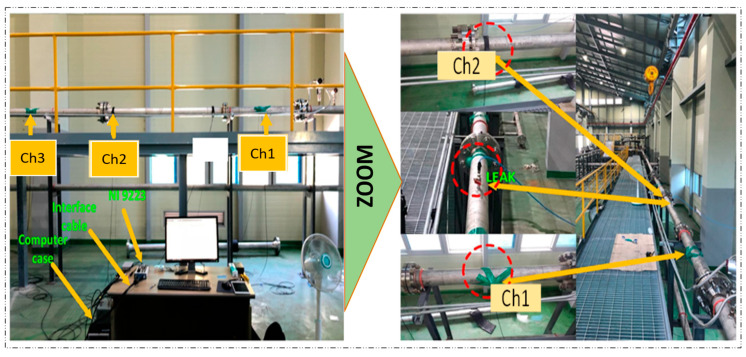
Experimental setup for sensor placement, leakage inducement, and data acquisition.

**Figure 6 sensors-23-03226-f006:**
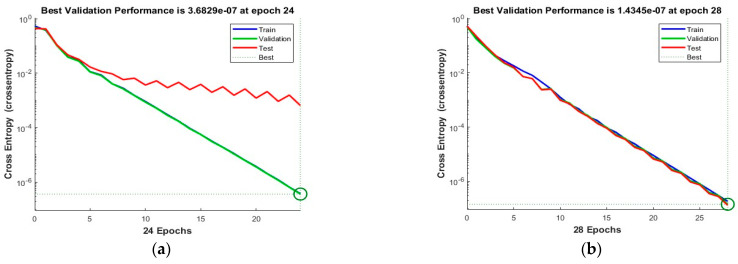
The convergence of the neural network for Dataset-1. (**a**) 10 Neurons and (**b**) 50 Neurons in the hidden layers.

**Figure 7 sensors-23-03226-f007:**
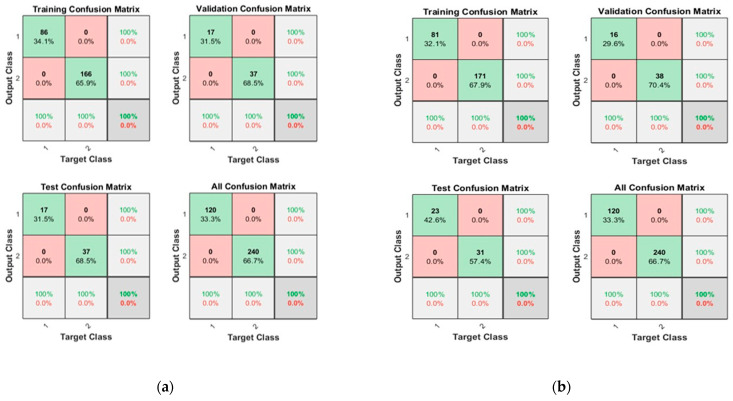
Confusion matrices of Dataset-1, where class label 1 shows Normal operating conditions and class label 2 shows the Leak operating conditions of the pipeline using (**a**) 10 Neurons and (**b**) 50 Neurons.

**Figure 8 sensors-23-03226-f008:**
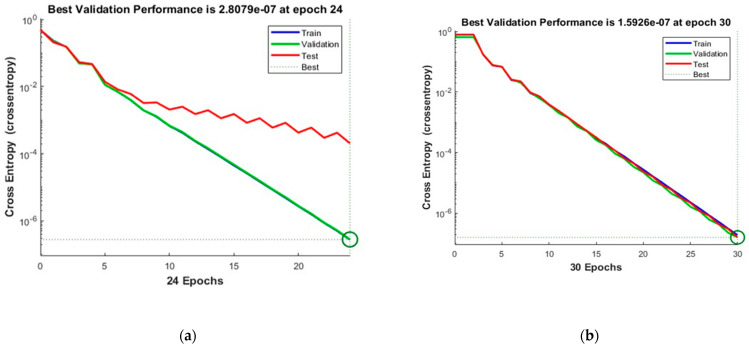
The convergence of the neural network for Dataset-2. (**a**) 10 Neurons and (**b**) 50 Neurons in the hidden layers.

**Figure 9 sensors-23-03226-f009:**
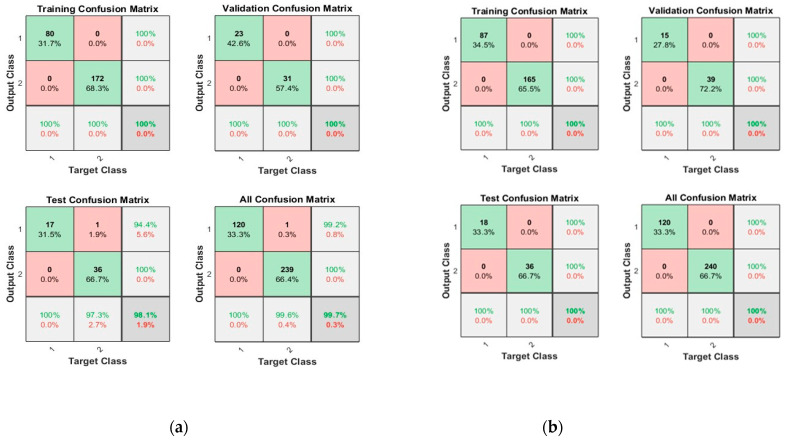
Confusion matrices of Dataset-2, where class label 1 shows the Normal operating conditions and class label 2 shows the Leak operating conditions of the pipeline using (**a**) 10 Neurons and (**b**) 50 Neurons.

**Figure 10 sensors-23-03226-f010:**
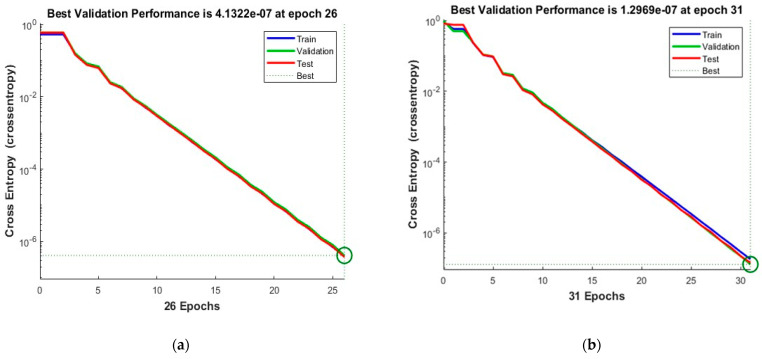
The convergence of the neural network for Dataset-3. (**a**) 10 Neurons and (**b**) 50 Neurons in the hidden layers.

**Figure 11 sensors-23-03226-f011:**
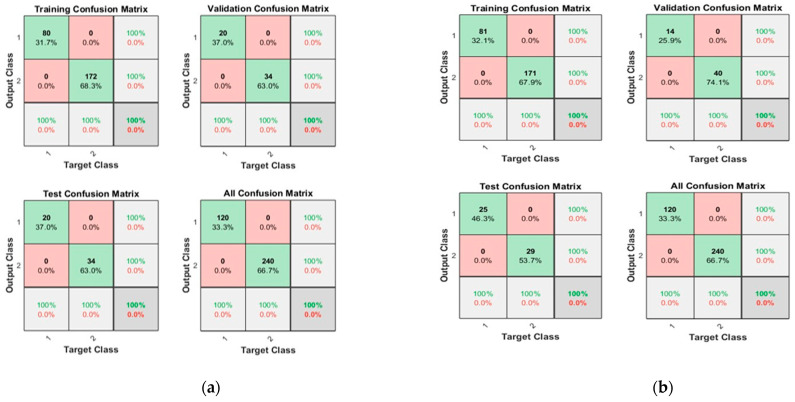
Confusion matrices of Dataset-3, where class label 1 shows the Normal operating conditions and class label 2 shows the Leak operating conditions of the pipeline using (**a**) 10 Neurons and (**b**) 50 Neurons.

**Figure 12 sensors-23-03226-f012:**
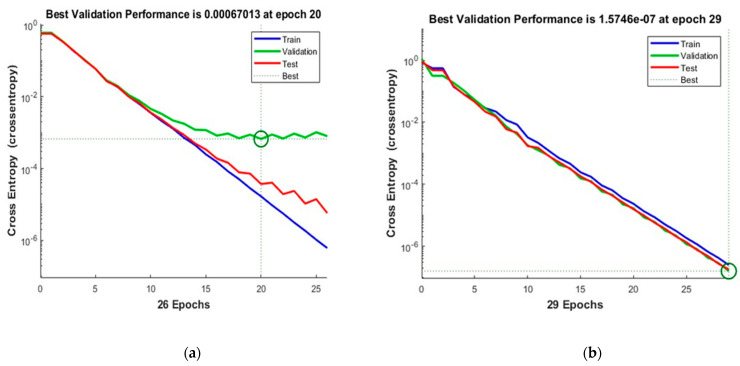
The convergence of the neural network for Dataset-4. (**a**) 10 Neurons and (**b**) 50 Neurons in the hidden layers.

**Figure 13 sensors-23-03226-f013:**
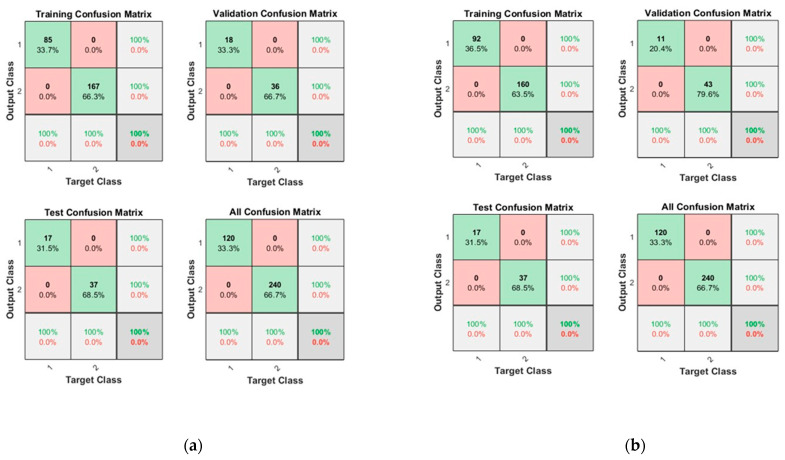
Confusion matrices of the Dataset-4, where class label 1 shows the Normal operating conditions and class label 2 shows the Leak operating conditions of the pipeline using (**a**) 10 Neurons and (**b**) 50 Neurons.

**Figure 14 sensors-23-03226-f014:**
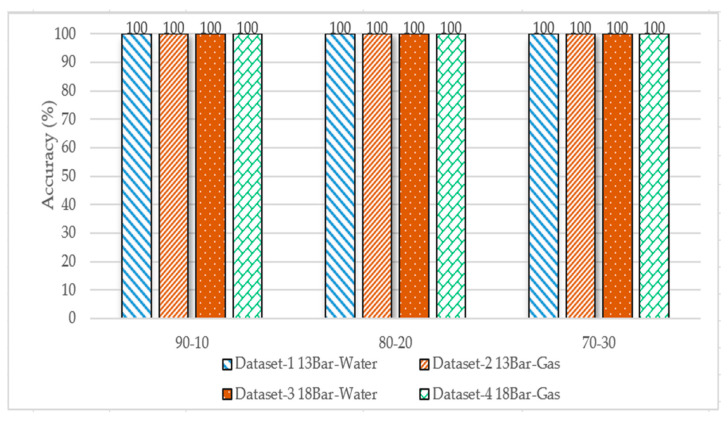
Performance comparison of KNN algorithm accuracy for the four datasets for different data splits.

**Figure 15 sensors-23-03226-f015:**
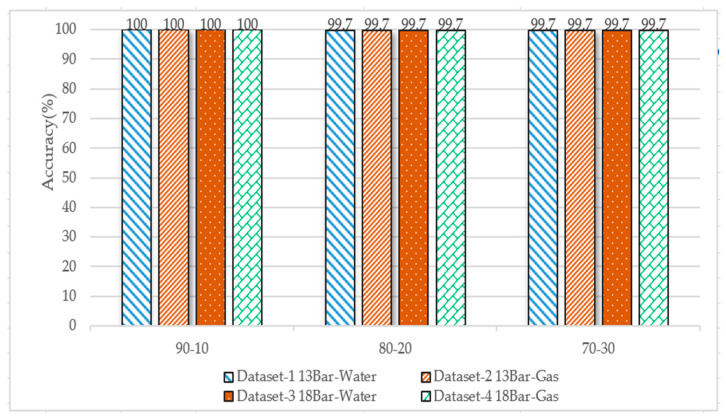
Performance comparison of random forest algorithm accuracy for the four datasets and for the different data splits (No. of Trees = 100; Gini index split).

**Figure 16 sensors-23-03226-f016:**
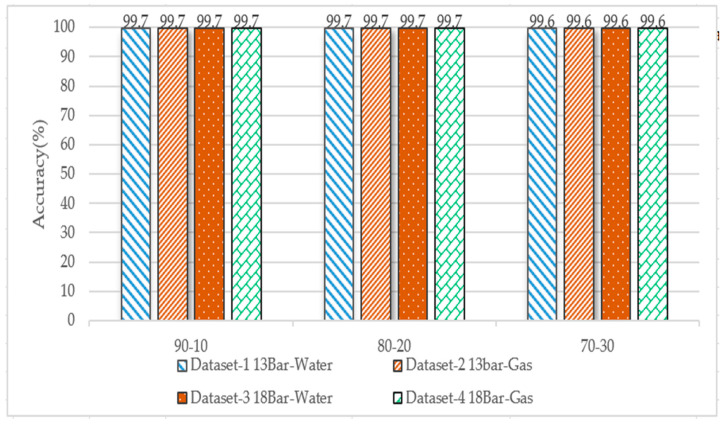
Performance comparison of decision tree algorithm accuracy for the four datasets for different data splits (Gini index split).

**Figure 17 sensors-23-03226-f017:**
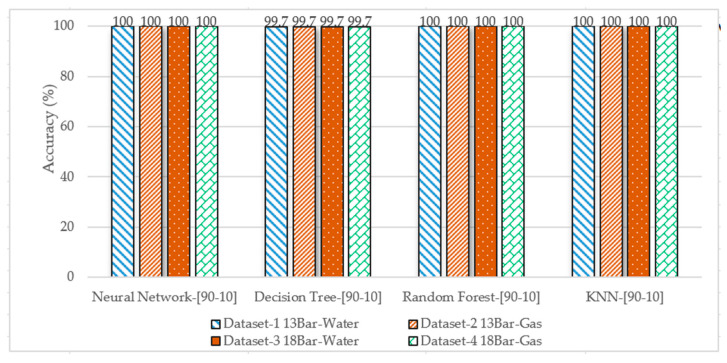
Performance comparison of the applied ML algorithms in terms of accuracy.

**Table 1 sensors-23-03226-t001:** Parameter for data acquisition.

S. No	Parameter	Value
1	AE sensor 1 location	2500 mm
2	AE sensor 2 location	1600 mm
3	AE sensor 3 location	0 mm
4	Peak Sensitivity	109 dB
5	Operational frequency range	80–200 kHz
6	Resonant frequency	75 kHz
7	Pipeline thickness	6.02 mm
8	Pipeline material	304 stainless steels
9	Pipeline outer diameter	114.3 mm

**Table 2 sensors-23-03226-t002:** Time domain features of AE channels [19].

Feature Name	Equation	Feature Name	Equation
Mean	$X_{m}=\frac{\sum_{n=1}^{N}x(n)}{N}$	Standard deviation	$X_{sd}=\sqrt{\frac{\sum_{n=1}^{N}{\left(x(n)-X_{m}\right)}^{2}}{N-1}}$
Root amplitude	$X_{root}={\left[\frac{\sum_{n=1}^{N}\sqrt{\left|x(n)\right|}}{N}\right]}^{2}$	Skewness	$X_{sk.}=\frac{\sum_{n=1}^{N}{\left(x(n)-X_{m}\right)}^{2}}{(N-1)X_{sd}^{3}}$
RMS	$X_{rms}=\sqrt{\frac{\sum_{n=1}^{N}{\left(x(n)\right)}^{2}}{N}}$	Kurtosis	$X_{ku}=\frac{\sum_{n=1}^{N}{\left(x\left(n\right)-X_{m}\right)}^{4}}{\left(N-1\right)X_{sd}^{4}}$
Impulse factor	$X_{impulse}=\frac{X_{peak}}{\frac{1}{N}\sum_{n=1}^{N}\left|x(n)\right|}$	Root value	$X_{root}={\left(\frac{\sum_{n=1}^{N}\sqrt{\left|x\left(n\right)\right|}}{N}\right)}^{2}$
Shape factor	$X_{shape}=\frac{X_{rms}}{\frac{1}{N}\sum_{n=1}^{N}\left|x(n)\right|}$	Crest factor	$X_{crest}=\frac{X_{peak}}{X_{rms}}$
Clearance factor	$X_{clearnace}=\frac{X_{peak}}{X_{root}}$		

**Table 3 sensors-23-03226-t003:** Frequency domain features of AE channels.

Feature Name	Equation	Feature Name	Equation
Mean Frequency	$P_{1}=\frac{\sum_{k=1}^{K}s\left(k\right)}{K}$	Fourth Moment of Frequency	$P8=\sqrt{\frac{\sum_{k=1}^{K}f_{k}^{4}\;s\left(k\right)}{\sum_{k=1}^{K}f_{k}^{2}\;s\left(k\right)}}$
Variance	$p2=\frac{\sum_{k=1}^{K}{\left(s\left(k\right)-P1\right)}^{2}}{K-1}$	Flattening Factor	$P9=\frac{\sum_{k=1}^{K}f_{k}^{2}s\left(k\right)}{\sqrt{\sum_{k}^{K}s\left(k\right)\;}\sum_{k=1}^{K}f_{k}^{4}\;s\left(k\right)}$
Skewness	$P3=\frac{\sum_{k=1}^{K}{\left(s\left(k\right)-p1\right)}^{3}}{k{\left(\sqrt{p2}\right)}^{2}}$	Coefficient of Variation of Centroid Frequency	$P10=\frac{P_{6}}{P_{5}}$
Spectral kurtosis	$P4=\frac{\sum_{k=1}^{K}{\left(\left(s\left(k\right)-p1\right)\right)}^{4}}{Kp_{2}^{2}}$	Skewness of Centroid Frequency	$P11=\frac{\sum_{k=1}^{K}{\left(f_{k}-P_{5}\right)}^{3}\;s\left(k\right)}{KP_{6}^{3}}$
Centroid frequency	$P5=X_{fc}=\frac{\sum_{k=1}^{K}f_{k}s\left(k\right)}{\sum_{k=1}^{K}s\left(k\right)}$	Kurtosis of Centroid Frequency	$P12=\frac{\sum_{k=1}^{K}{\left(f_{k}-P_{5}\right)}^{4}\;s\left(k\right)}{KP_{6}^{4}}$
Standard Deviation of Centroid Frequency	$P6=\sqrt{\frac{\sum_{k=1}^{K}{\left(f_{k}-P_{5}\right)}^{2\;}s\left(k\right)}{K}}$	Square Root of Centroid Frequency	$P13=\frac{\sum_{k=1}^{K}{\left(f_{k}-P_{5}\right)}^{1/2}\;s\left(k\right)}{K\sqrt{P_{6}}}$
Root means square frequency	$P7=X_{rmsf}=\sqrt{\frac{\sum_{k=1}^{K}f_{k}^{2}s\left(k\right)}{\sum_{k=1}^{K}s\left(k\right)}}$	Root Mean Square of Centroid Frequency Deviation	$P14=\sqrt{\frac{\sum_{k=1}^{K}f{\left(k-P_{5}\right)}^{2\;}s\left(k\right)}{\sum_{k=1}^{K}s\left(k\right)}}$

**Table 4 sensors-23-03226-t004:** Datasets acquisition configuration description.

Datasets	Pressure-Substance	Leak Pinhole Size	Acquisition Duration	Number of Feature Vector Samples (Normal/Leak)
Dataset-1	13 bar-Water	1 mm	6 min	120/240
Dataset-2	13 bar-Gas	0.5 mm	6 min	120/240
Dataset-3	18 bar-Water	0.7 mm	6 min	120/240
Dataset-4	18 bar-Gas	0.5 mm	6 min	120/240

## Data Availability

Data will be provided upon request.

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
