# Peer review of "Pipeline Leakage Detection Using Acoustic Emission and Machine Learning Algorithms"

_sensors, 2023, doi:10.3390/s23063226_

Round 1
Reviewer 1 Report
This paper has little innovation, and the methods proposed are too common, but the method is applied to the pipeline. The essence of the method is relatively ordinary and not innovative.
Reviewer 2 Report
This work utilized AE technique and some conventional machine learning methods to detect pipe leakage under different pressures. I believe the readers in Sensors will be very interested in the topic of this paper. However, the overall novelty of this paper is relatively low. The authors failed to propose new methods or strategies both in AE feature extraction and machine learning algorithms. The authors also failed to provide a thorough comparison of AE signals under the normal and leakage condition. Therefore, a major revision is required before publication.
1. Please specify the novelty and objective of this investigation at the end of the Introduction. Please do not show the major conclusions in Introduction.
2. In line 145, it is necessary to provide the operating frequency range of the AE sensor.
3. Missing unit in Figure 3. A detailed description of each graph should be provided in figure caption for readers’ better understanding.
4. The authors extracted many features in both time and frequency domains. However, the authors neglected many popular AE features such as count, energy, entropy and so on, which are widely used for damage identification in AE monitoring. The definitions of these AE parameters can be found in the following papers. I suggest the authors refer to the following papers and discuss these parameters extracted from signals under the normal and leakage conditions.
https://doi.org/10.1016/j.engfracmech.2020.107083
https://doi.org/10.1016/j.ijfatigue.2022.106860
https://doi.org/10.1038/s41524-021-00565-x
5. Four datasets were created in this work. Each dataset needs a clear description.
6. Can the four datasets merge into one? In the new dataset, the pressure and type of medium can also serve as the inputs of the machine learning models, which is more practical. Please discuss.
7. The dataset was too small for machine learning models, which is one possible reason for the very high accuracy (>99%) of each machine learning model.
8. In Line 234, I guess there is a typo error because the thickness should be much smaller than the outer diameter of the pipeline.
Reviewer 3 Report
This article proposes a machine learning-based platform for leakage detection using the AE sensor channel information. Several minor issues need to be addressed before publication.
1. As stated in Abstract, the leak pinhole size is also a detection object. However, this is not presented in the text anymore.
2. As stated in the paper, four datasets are used to train and test the models. But description of these four datasets is missing. How to obtain these four datasets? What is the differences between these four datasets?
3. As the Acoustic Emission approach is an important part of the proposed method, please introduce the basic principle and specific application methods in this study in detail.
4. Please reorganize the contents of Introduction. Some of the contents are repetitive, and it seems that the first paragraph of Introduction is redundant.
Reviewer 4 Report
1. Linguistic quality of this manuscript is not satisfactory. Many grammatical errors and poorly constructed sentences are found. Please proofread and improve the quality.
2. Please explain the differences between current work and those published works.
3. Problem statements and research gaps that motivated current study are not clearly explained. Further elaboration is needed to highlight the significance of current study.
4. Overview of proposed work should be explained before presenting the research contribution
5. Figure 2 looks distorted. Please improve the quality of this figure.
6. The employed machine learning models (i.e., neural network, KNN, decision tree and random forest) should be described. Please justify the selection of these machine learning models.
7. Equation for Precision is missing.
8. Confusion matrices presented in Figures 7, 9, 11 and 13 are too small. Need to enlarge these figures.
9. What are the output classes to be classified? Please explain.
10. FIgure 17 - The caption of this figure is incomplete.
11. More detailed discussion are needed to explain the performance differences between all selected ML models.
Reviewer 5 Report
Comments to the authors:
1. Kindly avoid 'we' in the entire article.
2. The research gap and novelty are not presented in the article.
3. Experimental setup image or photo needs to be added in the methodology section. Figure 5 can be supplemented with an actual photograph.
4. Time domain and frequency domain feature equations need a proper citation.
5. Figure 3 e is not available in the manuscript.
6. State the significance of the spikes in the power spectrum in Figure 3 e,f,g ad k,l,m
7. ANN architecture, algorithm, and Transfer function used for models are not clear.
8. KNN accuracy is 100%. It seems to overfit of data.
9. 90:10 is not universally followed data frame size for ML algorithms. The authors should go with 70:30.
10. A separate discussion section is mandatory.
11 Old references and books can be replaced with the latest articles.
Round 2
Reviewer 1 Report
After revision, the idea of this article is very clear, and the good part of this article is that it has practical application value.
Author Response
Thanks for your positive response.
Reviewer 2 Report
1. Kindly check the grammar and typo errors. For example, in Line 476, "however" should be "However".
2. The references should be renumbered when new references are added. See Line 82-86.
3. In Line 86, reference number [30,31] should not exist since this sentence corresponds to reference [29].
Reviewer 4 Report
The quality of the manuscript has been improved significantly after the extensive revision made by authors. In my opinion, this paper is suitable to be accepted for publication. Well done to all authors.
Author Response
Thanks for your positive response.
Reviewer 5 Report
Congrats to the authors.
Author Response
Thanks for your positive response.